# The Combined Treatment of Glutathione Sodium Salt and Ascorbic Acid for Preventing Contrast-Associated Acute Kidney Injury in ST-Elevation Myocardial Infarction Patients Undergoing Primary PCI: A Hypothesis to Be Validated

**DOI:** 10.3390/antiox12030773

**Published:** 2023-03-22

**Authors:** Alessio Arrivi, Giovanni Truscelli, Giacomo Pucci, Francesco Barillà, Roberto Carnevale, Cristina Nocella, Martina Sordi, Marcello Dominici, Gaetano Tanzilli, Enrico Mangieri

**Affiliations:** 1Interventional Cardiology Unit, “Santa Maria” University Hospital, 05100 Terni, Italy; 2Division of Cardiology, Bios Group, 00197 Rome, Italy; 3Unit of Internal Medicine, S. Maria University Hospital, 05100 Terni, Italy; 4Department of Medicine and Surgery, University of Perugia, 06123 Perugia, Italy; 5Department of Systems Medicine, University Tor Vergata, 00133 Rome, Italy; 6Department of Medical-Surgical Sciences and Biotechnologies Sapienza University, 04100 Latina, Italy; 7IRCCS Neuromed, Località Camerelle, 86077 Pozzilli, Italy; 8Department of Clinical, Internal Medicine, Anesthesiology and Cardiovascular Sciences, Sapienza University of Rome, 00161 Rome, Italy

**Keywords:** contrast-associated acute kidney injury, glutathione, ascorbic acid, STEMI, primary percutaneous coronary intervention

## Abstract

The occurrence of Contrast-Associated Acute Kidney Injury (CA-AKI) in patients with ST-Elevation Myocardial Infarction (STEMI) has a negative impact on the length of hospital stay and mortality. Reactive Oxygen Species (ROS) release, along with vasoconstriction and hypoperfusion, play a key role in its development. To date, there is still no validated prophylactic therapy for this disease. The use of antioxidants, based on experimental and clinical studies, looks promising. Taking into consideration previous literature, we speculate that an early, combined and prolonged intravenous administration of both Glutathione (GSH) and ascorbic acid in STEMI patients undergoing primary Percutaneous Coronary Intervention (pPCI) may be of value in counteracting the occurrence of CA-AKI. We aimed at evaluating this hypothesis by applying a multicenter research protocol, using a double-blind randomized, placebo-controlled trial design. The primary endpoint will be to test the efficacy of this combined antioxidant therapy in reducing the occurrence of renal damage, in patients with acute myocardial infarction treated with pPCI. Furthermore, we will investigate the effect of the study compounds on changes in oxidative stress markers and platelet activation levels through bio-humoral analyses.

## 1. Introduction

In-hospital Acute Kidney Injury (AKI) is an important acquired disease whose most frequent causes are the following: impaired renal perfusion, nephrotoxic drugs, and contrast media (CM) exposure [1]. The latter, named Contrast-Associated Acute Kidney Injury (CA-AKI), is present in more than 15% of ST-Elevation Myocardial Infarction (STEMI) patients who undergo primary percutaneous Coronary Intervention (p-PCI) [2,3]. It consists of a decline of renal function corresponding to an increase in serum creatinine over 25% or 44 μmol/L from baseline, within 3 days following the intravascular administration of iodinate CM agents [4]. It can occur in both patients with pre-existing renal disease and in those with normal renal function [5]; its development is associated with a prolonged length of hospital stay and increased mortality over the medium–long term [6,7]. Independent risk factors for the development of CA-AKI are the following: age, door to balloon time, troponin-T peak value, being of the female sex and contrast volume to eGFR ratio [8]. Its pathophysiology is still unknown and has been postulated to be related to renal ischemia, oxidative injury and direct toxicity to tubular epithelial cells [9,10,11]. To date, controversy still remains regarding the efficacy of prophylactic treatment in preventing the occurrence of CA-AKI. The proposed therapeutical strategies include hydration with sodium chloride or sodium bicarbonate, or the administration of N-Acetylcysteine (NAC), Angiotensin-Converting Enzyme Inhibitors (ACE-Is) (albeit with unconvincing clinical benefit) [12], statins or vitamin C [12,13,14]. Despite this, there is no consensus regarding what is the most effective intervention to prevent CA-AKI, and current guidelines still recommend the intravenous administration of isotonic saline or sodium bicarbonate in patients at increased risk for this disease [15].

The role of antioxidants and NAC is also debated. Indeed, although NAC enhances the effect of the endogenous vasodilator nitric oxide and counteracts antioxidant injury, a number of published studies provided inconclusive findings [16,17]. A possible explanation is that its scavenger activity is not fully capable of counteracting all intracellular reactive oxygen species’ (ROS) production and, therefore, is unable to prevent mitochondrial damage [18]. Furthermore, serious adverse effects, such as anaphylactoid reactions, were found in up to 8.2% of patients after the intravenous administration of this antioxidant [19].

The use of ascorbic acid has shown promising results in the literature, although to date there are still uncertainties about its dosage and route of administration, which impact its bioavailability [20].

## 2. The Study Hypothesis

We hypothesized that the intravenous administration of glutathione sodium salt (GSH) associated with ascorbic acid, just before p-PCI and during the following days, is effective in counteracting the onset of CA-AKI. This intervention could combine the demonstrated scavenging action of the two molecules with an increase in the GSH-mediated bioavailability of Nitric Oxide (NO). We are confident in believing that this integrated antioxidant/vasodilating action, along with an improved effective circulating blood volume, due to a better myocardium reperfusion, could have a role in preventing the occurrence of CA-AKI.

## 3. Antioxidant Properties of Glutathione Sodium Salt and Ascorbic Acid

Low-molecular-weight antioxidants, such as GSH, represent the first mechanism of protection against ROS, released due to oxidative bursts. GSH is a water-soluble tripeptide. Its antioxidant activity is associated with the sulfur chemical group, which includes potent reducing agents [21]. It acts directly by scavenging reactive oxygen and nitrogen species, or indirectly, by improving enzymatic activity as a cofactor [22,23]. Its oxidized state (GSSG) could be changed through the activities of isoenzyme glutathione reductase and Nicotinamide Adenine Dinucleotide Phosphate (NADPH). An increased GSSG-to-GSH ratio is suggestive of oxidative stress [21]. Oxidative stress levels are able to influence the total cellular GSH content, as well as the GSH/GSSG ratio, through a GSH negative-feedback loop [24]. The balance between synthesis and catabolism guides the intra- and extracellular content of GSH and its transport between the cytosol and the extracellular compartment [25]. GSH can cross the mitochondrial membrane, being stored the endoplasmic reticulum. Exposure to oxidative stress drives much of the cellularly synthesized GSH into the extracellular spaces across the plasma membrane [25]. Two enzymes (γ-glutamylcysteine synthetase (GCS) and GSH synthetase) catalyze the synthesis of GSH from glutamate, cysteine, and glycine [26]. This process is controlled by GCS activity, cysteine availability and GSH feedback inhibition.

Ascorbic acid is widely used as dietary supplement, but its antioxidant activity properties are well established to date [27,28]. Ascorbate is a valid scavenger of ROS (O_2_^•−^, H_2_O_2_, and ^•^OH). Its scavenging properties are related to its ability to form a stabilized radical; it acts mainly by donating an electron to harmful oxidizing radicals. This one-electron oxidation of AH ^−^ (fully reduced form) results in the production of the ascorbyl radical (A^•−^) (anion monodehydroascorbate); as a consequence, the reactive free radical is reduced [20,29]. Two-equivalent oxidation (double oxidation) would lead to the generation of the high-energy product pseudodehydroascorbate (A). This permits ascorbate to react with more reactive molecules, including the hydroxyl radical or the superoxide radical anion, thus avoiding cellular damage [27,28]. Of note, ascorbic acid significantly increases the level of GSH, without affecting the activity of enzymes associated with this molecule; indeed the action of ascorbic acid is primarily based on inducing plasma glutathione peroxidase (GSH-Px) and glutathione reductase (GSSG-R) activity. Therefore, from a biochemical point of view, we can state that the kinetics of ascorbate reflect those of a co-antioxidant, which favors other scavengers’ renewal; Indeed, the activity of multiple transcription factors (Nrf2, Ref-1, AP-1) is stimulated by ascorbate, thus allowing the expression of genes that encode antioxidant proteins [27,28]. As a reducing agent, ascorbic acid could reduce ferric iron to ferrous iron, thus facilitating the generation of ROS. However, this hypothetical pro-oxidant action has been demonstrated mostly in vitro, as to date the in vivo data are inconclusive [20,30,31,32].

## 4. Pharmacokinetic Properties of Exogenous Glutathione and Ascorbic Acid

The bioavailability of GSH from oral administration is rather low, due to its degradation by the γ-Glutamyl Transpeptidase (GGT) enzyme. Instead, after intravenous infusion of GSH at a dose of 2 g/m^2^ in healthy volunteers, the plasma total glutathione concentration increased until 823 ± 326 µmol/L [33]. The volume of distribution of GSH was calculated to be 176 ± 107 mL/kg, while the plasma half-life was 14.1 ± 9.2 min. The concentration of cysteine in the plasma increased to 114 ± 45 µmol/L after infusion. Despite this, the total plasma concentration of cysteine, cystine and mixed disulfides decreased, thus indicating increased passage of cysteine into the cell’s compartment. Urinary excretion of GSH and cyst(e)ine showed increases of 300% and 10%, respectively, in the 90 min following the infusion [33]. These data indicate that the intravenous administration of GSH is able to increase the concentration of sulfhydryl compounds in the urinary system, and, therefore, improve the cellular availability of cysteine. The high intracellular concentration of cysteine justifies the protection against xenobiotics, as it results, directly or indirectly, in an increase of GSH synthesis [33].

Ascorbic acid is rapidly absorbed from the gastrointestinal tract by a saturable, sodium-dependent, active transport mechanism. At low concentrations, vitamin C is absorbed by active transport; at high concentrations, passive diffusion predominates. In the cases of doses between 30 and 180 mg/day (intake of vitamin C with food), the absorbed amount varies between 70 and 90%; in the case of high doses (>1 g/day), as occurs, for example, with multivitamin supplements, the amount of the absorbed vitamin decreases by up to 50%, and can be even lower [34]. In fact, by increasing the dose of ascorbic acid from 200 mg to 2500 mg (from 1.1 to 14.2 mmol), its plasma concentration under steady state conditions increases to a much lesser extent, approximately from 12 to 15 mg/L (from 68.1 to 85.2 mums) [35]. Furthermore, if the doses are very high, most of the vitamin remains inside the intestinal lumen, not absorbed, causing gastrointestinal disturbances and diarrhea [34]. To overcome the concern related to reduced bioavailability from intestinal sequestration, parenteral administration is used. A second mechanism of the homeostasis of ascorbic acid is represented by renal secretion. In the case of high doses, the tubular reabsorption system becomes saturated, and the amount of ascorbic acid excreted in the urine increases exponentially. Conversely, when the intake of ascorbic acid is minimal, the body tends to preserve its stocks as much as possible, minimizing the amount that can be eliminated. The renal threshold is 1.5 mg/dL; daily doses exceeding 100 mg are eliminated in the urine with a slight diuretic effect [34]. It is found in the urine both unchanged and as dehydroascorbic acid, as well as in the form of metabolites (2,3-diketogulonic acid and oxalic acid). The concentration of ascorbic acid in the blood was found to be higher in women than in men. This difference would seem to depend (10–30%) on the different distributions of lean mass and fat mass between the two sexes (men have a higher lean mass index and a lower fat mass index than women) [36]. The concentration of ascorbic acid in the blood of smokers is lower (by about 40%) than that observed in non-smokers. The smoke of a cigarette is able to neutralize the ascorbic acid content of an orange (about 25 mg of vitamin C). Furthermore, in smokers, nicotine appears to reduce the gastrointestinal absorption capacity of ascorbic acid and increase its metabolism (about 35%), thus increasing the daily requirement of the vitamin [37,38].

## 5. Evaluation of the Hypothesis

According to our hypothesis, we believe that it is reasonable that (1) ischemia-reperfusion injury (IRI) during STEMI could substantially contribute to renal damage through ROS liberation, vasoconstriction and hypoperfusion [9,11] (Figure 1). (2) GSH [39,40] and ascorbic acid [20,41,42] could counteract this process both through their scavenging action against some of these free radicals, such as hydrogen peroxide (H_2_O)_2_, and through enhanced vasodilatation, due to the documented favorable GSH effect on NO availability [39,43]. Furthermore, the reduction in myocardial cell damage could improve renal perfusion, which, in turn, can reduce the likelihood of secondary renal ischemia [44] (Figure 2). Based on previous results [39,40,41,42,43,45], it is plausible to hypothesize that a combination of the two molecules, using the same modality of administration (such as early and prolonged intravenous administration over time), could be effective. To confirm this hypothesis, after obtaining all institutional authorizations, we will apply a multicenter research protocol, using a double-blind randomized, placebo-controlled trial design. The primary endpoint will be to test the efficacy of this combined antioxidant therapy, in terms of effectively reducing the occurrence of renal damage in STEMI patients. We plan to enroll at least 400 patients with STEMI of an age > 18 years, and of both genders, recruited from three hub centers in the middle of Italy: Department of Clinical, Internal Medicine, Anesthesiology and Cardiovascular Sciences, Sapienza University of Rome; Department of Systems Medicine, University Tor Vergata, Rome; and Interventional Cardiology Unit, “Santa Maria” University Hospital, Terni. Based on our previous results [43], considering an estimated incidence of CA-AKI following pPCI of about 20%, and predicting that GSH infusion would reduce the incidence of CA-AKI in the entire population by about 9%, we assumed that the enrolment of 400 patients would provide adequate power (70%) to rule out Type I errors, with an alpha level of 0.05%. Patients will be excluded for any of the following reasons: symptom duration >12 h, rescue PCI, cardiogenic shock, left main disease, prior myocardial infarction, in-stent thrombosis, saphenous venous graft occlusion, known AKI, end-stage kidney disease requiring dialysis, intravascular administration of CM in the few days preceding the pPCI, the need for early re-administration of CM in the three days following to pPCI, acute infection, treatment with systemic corticosteroids or oral anticoagulants, the presence of malignancies, known allergy to the CM, and lack of consent to participate. An individual not involved in the study will assign codes (using a computer-generated random sequence) to the study treatment, with a random allocation of patients to the treatment group or placebo. After giving their informed consent, all eligible patients will be, therefore, randomly assigned in a 1:1 manner to the treatment arm (200 pts) or to the placebo one (200 pts). The treatment will consist of an intravenous infusion of glutathione sodium salt (GSS, 2500 mg/25 mL, Biomedica Foscama Group, Rome, Italy) plus ascorbic acid (1 gr of vitamin C Vita (Sanofi-Aventis, Milan, Italy) for 10 min, prior to p-PCI. The placebo group will receive the same amount of a NaCl 0.9% saline solution. The solutions will appear similar to ensure blinding. Study participants, investigators and the laboratory staff will remain unaware of study-treatment allocation until the statistical analysis is performed, by an independent researcher not involved in the study. Patients will undergo p-PCI according to the standard protocols. The use of thrombus aspiration will be left to the discretion of the treating physician; instead, the type of contrast agent used to perform angiography, in order to avoid confounding effects, will be identical for all the catheterization laboratories involved in the study, comprising a nonionic, low or iso-osmolar contrast agent. All patients will be treated with a drug-eluting stent implant. After interventions, GSS + ascorbic acid will be infused at the same doses at 24, 48 and 72 h. In order to verify if the administration of the two molecules will decrease the occurrence of CA-AKI, we will analyze changes in serum creatinine concentrations between the experimental and the placebo groups, at baseline, as well as at 24, 48 and 72 h after p-PCI. All measurements will be performed in the same hospital laboratory, where each patient will be treated and allowed to recover. CI-AKI will be defined as an increase in serum creatinine over 25% or 44 μmol/L from baseline, within 3 days from the intravascular administration of the iodinated CM agent. For each procedure, the anthropometric and clinical data of each patient will be collected, and the total volume of iodinated CM used will be recorded. Furthermore, we will investigate the effect of the study compounds on changes in oxidative stress marker and platelet activation levels, such as *Platelet sP-selectin*, *H_2_O_2_* production, *sNox2-dp*, *Nitric Oxide*, *Platelet Aggregation* and *Thrombus Formation*. All blood samples will be drawn from a peripheral vein, before the start of procedure and after pPCI, in all patients, and then will be collected into tubes without anticoagulant, or with 3.8% sodium citrate, lithium heparin and EDTA, and centrifuged at 300× g for 10 min to obtain the supernatant. All plasma and serum aliquots will be stored at −80 °C in appropriate cuvettes until assayed. Markers of oxidative stress and antioxidant system will be analyzed in serum samples collected before p-PCI, as well as samples taken 24 h, 48 h and 3 days after p-PCI. Soluble P-selectin (sP-selectin) levels will be measured with a commercial immunoassay. The values will be expressed in ng/mL. The Hydrogen Peroxide (H_2_O_2_) will be measured by using a colorimetric assay. A standard curve of H_2_O_2_ (0–200 μM) will performed for each assay. The reaction product will be measured spectrophotometrically at 450 nm, and expressed as μM. Nox2 activity will be measured in serum and platelets, and expressed as sNox2-dp, with an ELISA method. Values will be expressed as pg/mL. NO production will be evaluated by a colorimetric assay kit (Abcam, Cambridge, UK), in order to determine the metabolites of nitric oxide (NO) (nitrites and nitrates, NOx) in a sample under stirring conditions for 10 min at 37 °C. Values will be expressed as μM. Platelet aggregation will be performed in PRP samples and monitored using a ChronoLog model 700 light-transmitting aggregometer. Samples will be activated with collagen (2 μg/mL, Mascia Brunelli, Milan, Italy) at 37 °C and aggregation will be recorded for 10 min. Values will be expressed as the percentage (%) of maximal aggregation. Thrombus growth under flow conditions will be measured by a thrombus formation analysis system (T-TAS^®^01 apparatus, Fujimori Kogyo Co., Ltd., Tokyo, Japan) on PL-chips (26 collagen-coated microcapillaries). The growth, intensity, and stability of the formation of platelet clots will be measured by the time needed to reach the occlusion pressure (occlusion time), and the area under the flow-pressure curve (AUC) parameter will be studied.

## 6. Discussion

AKI’s occurrence in the acute myocardial infarction setting is a complex process, in which not only the exposure to the CM itself, but also the hemodynamic changes, resulting from the acute left ventricular ischemic injury, become relevant. The decrease in cardiac output, depending on the type and extent of myocardial ischemia, the hydration status, as well as procedural factors (such as vascular access, CM volume and type) and possible comorbidities, are all relevant in its pathogenesis. The interplay between cardiac, volumetric and clinical characteristics leads to a consensual reduction of the glomerular filtration rate (cardio-renal mechanism) [46]. For the above reasons, we prefer referring to the term CA-AKI in this context, emphasizing the association of renal damage with other factors [4]. The oxidative stress induced by the release of ROS due to the acute event, and the vasoconstriction caused by activation of the sympathetic nervous system, have a key role in the onset of AKI [20,46]. ROS may carry out both direct nephron and endothelial damage, further amplifying renal parenchymal hypoxia throughout the inflammatory burst, occurring early after exposure to the CM and lasting for hours [47,48]. The latter is further heightened by myocardial reperfusion injury, which is a STEMI-related process that begins during the early phase of reperfusion, and continues for hours or days with apoptosis and inflammation [49]. The oxidative stress [50] following enhanced ROS production and reduced GSH availability act as the main actors of this phenomenon [45]. ROS induce the opening of Mitochondrial Permeability Transition Pore (MPTP) and exert a chemotactic action on white blood cells [51,52,53]. Specifically, the oxidative stress stimulates a strong free-radical activity that induces an important inflammatory response, with leukocytes’ adherence at the level of reperfused areas, as well as their further infiltration into the wounded tissue [54,55]. Major risk factors for AKI development are sepsis, hypovolemia, Chronic Kidney Disease (CKD) and diabetes mellitus [56]. The black race has an increased risk of AKI occurrence, which seem to be related to a direct interaction between genetic, as well as clinical and social class, factors [57]. Its pathophysiology consists of a complex multifactorial process [58]. Ischemia is largely considered the most frequent cause of AKI. It leads to microvascular impairment and to tubular injury. The latter, generally engaging the proximal tubular cells, may cause both lethal and sublethal cell injury, resulting in the cytoskeleton’s disruption. This, in turn, generates the following: loss of cell polarity, the shedding of cells, and cellular debris. The result is an altered vectorial transport, tubular obstruction and back-leak [58,59]. Of paramount importance in this microscopic operation is the activation of the inflammation cascade, resulting from factors released by damaged tubules, as well as the adhesion of white blood cells. The interplay between inflammation, microvascular and tubular injury leads to impaired renal function [58,59]. Ascorbic acid has a well-demonstrated scavenging action on ROS, such as O_2_^•−^, H_2_O_2_, and ^•^OH, and may form a part of effective prophylactic pharmacological regimens against contrast-induced nephropathy [20]. Indeed, a meta-analysis of nine, randomized, controlled trials provided consistence evidence of effective nephroprotection of ascorbic acid towards CA-AKI, with a 33% lower risk of kidney injury in treated patients [20]. A prospective, single-center, placebo-controlled, randomized study clearly showed that ascorbic acid infusion in patients undergoing PCI was able to improve myocardium reperfusion via the inhibition of oxidative stress, thus leading to a higher increase of Left Ventricular Ejection Fraction (LVEF) in the ascorbic acid-treated group when compared with the placebo one [41]. Moreover, the prophylactic oral administration of ascorbic acid, prior to coronary angiography and/or intervention, has been shown to protect against contrast-mediated nephropathy in high-risk patients [42]. However, to date, perplexities persist regarding the optimal dosage and bioavailability of the compound. Indeed, ascorbate scavenging is a dose-dependent phenomenon, requiring intravenous administration to react with superoxide anion radicals [20]. On the other hand, in agreement with our previous hypothesis [21], we demonstrated, within the context of a randomized clinical trial [39,40], that an early (immediately before p-PCI) and prolonged (up to 72 h after angioplasty) administration of GSH in patients with STEMI was able to significantly reduce H_2_O_2_ production, and also to promote a sustained increase of serum H_2_O_2_ Breakdown Activity (HBA) and NO bioavailability. In addition, we showed a progressive, significant decrease of serum cardiac troponin T (cTpT) release during the 5 days of reperfusion, in the GSH-treated patients when compared with the control group, resulting in a 21% reduction of myocardial damage [39], and in a significant shorter length of hospital stay [60]. Furthermore, in a prespecified randomized subgroup analysis of the same trial (GSH2014) [43], we observed a less steep rise in serum NO bioavailability in patients who developed CA-AKI than in the control group, thus demonstrating a possible pivotal role of NO depletion in determining contrast-mediated nephropathy. On these premises, we believe that the early and combined intravenous administration of the two antioxidants, starting before p-PCI and continuing up to 72 h from the index procedure, could counteract the development of CA-AKI through an enhanced scavenging action on the free radicals, a sustained vasodilation due to an increase in the NO bioavailability, and reduced renal ischemia, resulting from a better myocardial reperfusion [39,40,41,43] (Figure 2). The application of this new prophylactic treatment could be of value in the STEMI setting, where hypoperfusion, as a consequence of left ventricular dysfunction, as well as the reduction of effective blood volume, both contribute substantially to renal damage [44,61]. It must also be considered that during an AMI, and the consequent need for timely pPCI, it is not always possible to proceed with preventive hydration of the patient, thus losing the benefit of an adequate pre-treatment with saline solution. The integrated antioxidant treatment proposed (GSS plus ascorbic acid) trough parenteral infusion has the potential to intervene, promptly, in the pathogenesis of acute kidney injury, thanks to the achievement of a rapid steady state, and its supplementation once a day following the pPCI could decrease renal cell/tubule ROS-mediated damage [62,63] and improve renal function [64]. Kidney injury is, in fact, a slow process of death/apoptosis that takes place in hours/days following the acute event; therefore, a single dose of antioxidant, considering the pharmacokinetic properties of the two molecules reported above [33,34], may not be sufficient to fully counteract the kidney injury. Conversely, repeated post-event administration has the potential to better prevent the oxidative cascade leading to cellular death. In addition, the action of ascorbate, able to increase the serum levels of GSH [27,28], could further leading to a long-lasting scavenging action. Moreover, the use of these compounds, at the proposed doses, is free from significant side effects, which could invalidate their use [33,65]. The last-mentioned advantage is remarkable, allowing for a safe application during daily clinical practice. Preparation for parenteral infusion is simple, requiring only the utilization of a peripheral venous cannula. This ease of use fits with the context of an emergency procedure, such as the pPCI for STEMI, not interfering with the times and methods of the intervention. Last but not least, the costs of the aforementioned antioxidants are low [66,67], thus allowing for their easy purchase in most hospitals. This would make it possible to contain healthcare costs, considering the already high expenditure for acute myocardial infarction and the prolongation of hospital stays due to contrast-associated nephropathy [68,69]. The correlation of serum renal function data with bio-humoral oxidative stress data will allow us to biochemically quantify the extent and timing of the post-treatment reduction of the oxidative state, further validating the efficacy of the prophylactic combination therapy. We are confident that our study hypothesis can significantly contribute to research on a disease (such as CA-AKI) for which a validated therapy does not yet exist. The results of this investigation will also offer hints for possible future associations concerning the treatment of other ROS-related disorders.

## 7. Concluding Remarks

CA-AKI following STEMI is a worrying clinical condition, associated with adverse outcomes, due to both cardiac and renal function impairment. To date, there is still no specific treatment once it is confirmed, and prevention is the best option. However, the maintenance of an adequate blood volume with sodium chloride or sodium bicarbonate may not be sufficient or timely in the context of acute left ventricle damage following AMI, and reperfusion injury may get worse, through ROS-induced cell deterioration. The synthetic antioxidant supplementation proposed, using both GSS and ascorbic acid, based on previous scientific reports, could counteract this process, improving the patients’ outcomes.

## Figures and Tables

**Figure 1 antioxidants-12-00773-f001:**
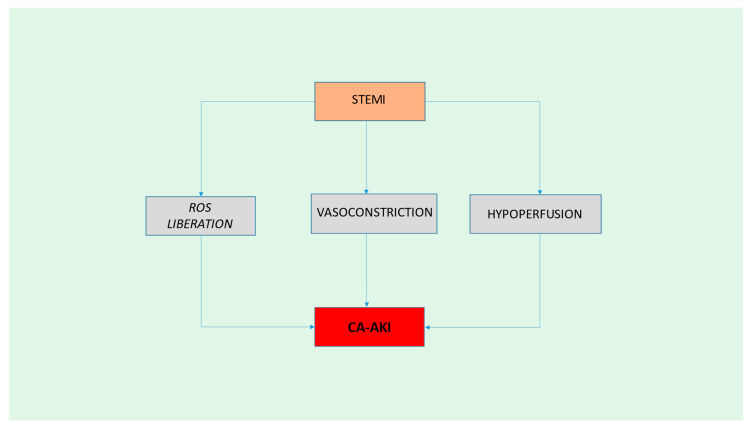
Pathophysiology of CA-AKI. STEMI: ST-elevation myocardial infarction; ROS: Reactive Oxygen Species; CA-AKI: Contrast-Associated Acute Kidney Injury.

**Figure 2 antioxidants-12-00773-f002:**
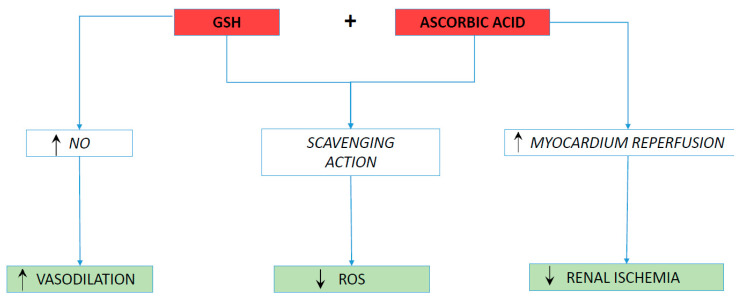
Graphical summary of our hypothesis. GSH: Glutathione; NO: nitric oxide; ROS: Reactive Oxygen Species.

## Data Availability

Data sharing is not applicable to this article.

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
