# Peer review of "The Combined Treatment of Glutathione Sodium Salt and Ascorbic Acid for Preventing Contrast-Associated Acute Kidney Injury in ST-Elevation Myocardial Infarction Patients Undergoing Primary PCI: A Hypothesis to Be Validated"

_antioxidants, 2023, doi:10.3390/antiox12030773_

Round 1
Reviewer 1 Report
I have read with great attention and interest the paper which opens up the possibility of preventing CI-AKI.
The topic is very interesting but appears to be not treated in a a nephrological point of view especially as regards the materials and methods in which reference is made to the eGFR in situations of non-steady state of creatinine in the case of AKI.
Furthermore, creatinine can vary not only in the course of AKI induced by a CM but also in the course of AKI associated with the use of a CM, a fact which is not well explained in the paper (AKI post use of a CM or due to a cardiorenal syndrome after the CM use).
There are inaccuracies in the units of measurement and their writing in the text.
Line 39 umol/L is correct, please check it.
Line49-53: it is not clear the sentence, it seems that ACEi can be protective against CI-AKI, please better explain.
Line 96 typing error.
Line 99 electron
Line 121, please check the volume of distribution
Line 177 gender and not sex
Line 180-187: the sentence is not clear and the terms used are not correct, please check ( known acute renal failure => AKI? AKD?.....)
Line 192 glutathione
Line 198-201: the choice of contrast media cannot be from the cardiologist but alla patients have to receive the same type of contrast to reduce the bias, please explain better your choice.
Line 204: the use of eGFR is not correct as the patient is affected by AKI the creatinine is not at the steady state and the formula cannot be use. Please explain this point better and change it.
210 iodinated CM agent, please correct the wrong iodinate
The discussion should focus more on the AKI pathophysiology and recall the literature in this area with particular reference to the risk of AKI, the Kidney damage and its pathophysiology with the possible role of the two molecules in the prevention of AKI.
Author Response
I have read with great attention and interest the paper which opens up the possibility of preventing CI-AKI.
The topic is very interesting but appears to be not treated in a nephrological point of view especially as regards the materials and methods in which reference is made to the eGFR in situations of non-steady state of creatinine in the case of AKI.
Thank You for Your comment. We agree with You and in order to avoid confusion we removed from the text the sentence related to eGFR.
Furthermore, creatinine can vary not only in the course of AKI induced by a CM but also in the course of AKI associated with the use of a CM, a fact which is not well explained in the paper (AKI post use of a CM or due to a cardiorenal syndrome after the CM use).
Thank You for Your observation. Indeed, the occurrence of AKI in the AMI setting results from multiple mechanisms acting synergistically. Hemodynamic abnormalities resulting from reduced cardiac output and venous return consequently lead to a decrease in glomerular filtration rate (GFR) [cardio-renal mechanism]. In addition, patients with AMI experience activation of the sympathetic nervous system and the renin-angiotensin-aldosterone system (RAAS), which can lead to vasoconstriction and exacerbate kidney damage. Furthermore also the activation of the inflammatory/oxidative response has deleterious effects on renal function by direct tubular and vascular endothelial injury. For the reasons mentioned above, we believe that the mechanism is cardiorenal, with a prevalent inflammatory/oxidative component. Therefore we replaced the wording CI-AKI with CA-AKI both in the title and in the text. Moreover we clarified this point in the discussion, please see at page 8, lines 307-315: “…AKI’s occurrence in the acute myocardial infarction setting, is a complex process in which not only the exposure to the CM itself but also the hemodynamic changes, resulting from the acute left ventricular ischemic injury, become relevant. The decrease in cardiac output depending on the type and extent of myocardial ischemia, the hydration status, as well as procedural factors (such as vascular access, CM volume and type) and possible comorbidities are all relevant in its pathogenesis. The interplay between cardiac, volumetric and clinical characteristics leads to a consensual reduction of the glomerular filtration rate (cardio-renal mechanism) [47]. For the above reasons we prefer referring to the term CA-AKI in this context, emphasizing the association of renal damage to other factors [4]. …”.
There are inaccuracies in the units of measurement and their writing in the text.
Line 39 umol/L is correct, please check it.
Thank you. Amended
Line 49-53: it is not clear the sentence, it seems that ACEi can be protective against CI-AKI, please better explain.
Thank you. Indeed ACE-Is act by inhibiting the renin-angiotensin-aldosterone system (RAAS), specifically the conversion of angiotensin-I to angiotensin-II , thereby causing vasodilatation of the efferent renal arterioles and thus decreasing the intraglomerular pressures [Dzau VJ. Mechanism of action of angiotensin-converting enzyme (ACE) inhibitors in hypertension and heart failure. Role of plasma versus tissue ACE. Drugs. 1990;39(2):11–16.]. They are thus called reno-protective because of this effect. On the other hand, ACE-Is also inhibit the formation of transforming growth factor beta-1 (TGF-β1) directly or through the inhibition of angiotensin-II [Shin GT, Kim SJ, Ma KA, Kim HS, Kim D. ACE inhibitors attenuate expression of renal transforming growth factor-β1 in humans. The American Journal of Kidney Diseases. 2000;36(5):894–902] which may promote proximal tubular cell injury. Currently there is no convincing renal benefit associated with continuing ACE-Is prior to angiography and the real possibility of harm [Kalyesubula R, Bagasha P, Perazella MA. ACE-I/ARB therapy prior to contrast exposure: what should the clinician do? Biomed Res Int. 2014;2014:423848.]. We clarified this in the introduction. Please see page 2 lines 55-56: “…albeit with unconvincing clinical benefit…”
Line 96 typing error.
Thank You. Amended
Line 99 electron
Thank You. Amended
Line 121, please check the volume of distribution
Thank You. We checked that the volume of distribution of exogenous glutathione is 176 +/- 107 ml kg-1. [Aebi S, Assereto R, Lauterburg BH. High-dose intravenous glutathione in man. Pharmacokinetics and effects on cyst(e)ine in plasma and urine. Eur J Clin Invest. 1991 Feb;21(1):103-10].
Line 177 gender and not sex
Thank You. Amended
Line 180-187: the sentence is not clear and the terms used are not correct, please check (known acute renal failure => AKI? AKD?.....)
Thank You. Amended
Line 192 glutathione
Thank You. Amended
Line 198-201: the choice of contrast media cannot be from the cardiologist but all patients have to receive the same type of contrast to reduce the bias, please explain better your choice.
Thank You for Your suggestion. We agree with you that, in order to avoid bias, the contrast medium used must be the same for all three centers involved in the trial. Therefore we changed to: “…instead the type of contrast agent used to perform angiography, in order to avoid confounding effects, will be identical for all the catheterization laboratories involved in study: the same nonionic, low or iso-osmolar contrast agent…”. Please see at page 5, lines 232-235.
Line 204: the use of eGFR is not correct as the patient is affected by AKI the creatinine is not at the steady state and the formula cannot be use. Please explain this point better and change it.
Thank You for pointing out this aspect. We removed the sentence related to eGFR; please see at page 5, lines 239-240: “…In order to verify if the administration of the two molecules will decrease the occurrence of CA-AKI, we will analyze changes in serum creatinine concentrations between the experimental and the placebo groups baseline, at 24, 48 and 72 hours after p-PCI.…”
210 iodinated CM agent, please correct the wrong iodinate
Thank You. Amended
The discussion should focus more on the AKI pathophysiology and recall the literature in this area with particular reference to the risk of AKI, the Kidney damage and its pathophysiology with the possible role of the two molecules in the prevention of AKI.
Thank You for Your suggestion. We implemented the discussion on this topic. Please see at page 8, lines 335-336 and page 9 lines 338-347: “…Major risk factors for AKI development are sepsis, hypovolaemia, chronic kidney dis-ease (CKD) and diabetes mellitus [57]. Its pathophysiology consists of a complex multifactorial process [59]. Ischemia is largely considered the most frequent cause. It leads to microvascular impairment and to tubular injury. The latter, generally engaging the proximal tubular cells, may cause both lethal and sublethal cell injury with following cytoskeleton’s disruption. This in turn generates: loss of cell polarity, shed cells and cellular debris. The result is an altered vectorial transport, tubular obstruction and backleak [59,60]. Of paramount importance in this microscopic operation is the activation of the inflammation cascade from factors released by damaged tubules and adhesion of white blood cells. The interplay between inflammation, microvascular and tubular injury leads to impaired renal function [59,60]…” and page 9, lines 375-380: “…On these premises, we believe that the early and combined intravenous administration of the two antioxidants, starting before p-PCI and up to 72 hours from the index procedure, could counteract the development of CA-AKI through an enhanced scavenging action on the free radicals, a sustained vasodilation due to an increase in the NO bioavailability and reduced renal ischemia resulting from a better myocardium reperfusion [40-42, 44] [Figure 2]…”
Author Response
General comments
This manuscript describes the hypothesis of the combined and prolonged intravenous administration of both glutathione and ascorbic acid in ST-elevation myocardial infarction patients undergoing primary percutaneous coronary intervention. The topic addressed is interesting. I think however that there are a few improvements that should be made before publication.
Specific comments
- In the title, the authors should not use abbreviations.
Thank You. Amended in: “ST-elevation myocardial infarction”
- In the introduction, what are the top 2 causes of in-hospital AKI?
Thank You. Indeed the two most common causes of hospital-acquired AKI are impaired renal perfusion and nephrotoxic medications [McCullough PA, Wolyn R, Rocher LL, Levin RN, O'Neill WW. Acute renal failure after coronary intervention: incidence, risk factors, and relationship to mortality. Am J Med. 1997 Nov;103(5):368-75]. We pointed out this data in the introduction; please see at page 1 lines 37-39: “…In-hospital Acute Kidney Injury (AKI) is an important acquired disease whose most frequent causes are: impaired renal perfusion, nephrotoxic drugs, and contrast media (CM) exposure [1]…”
- In lines 39-40, is the “44 μmol x L-1” same as “44 μmol/L”?
Thank You. Amended.
- In lines 49-51, a reference is needed.
Thank You. We included the following references: [Pistolesi V, Regolisti G, Morabito S, Gandolfini I, Corrado S, Piotti G, Fiaccadori E. Contrast medium induced acute kidney injury: a narrative review. J Nephrol. 2018 Dec;31(6):797-812. Kusirisin P, Chattipakorn SC, Chattipakorn N. Contrast-induced nephropathy and oxidative stress: mechanistic insights for better interventional approaches. J Transl Med. 2020 Oct 20;18(1):400. Kalyesubula R, Bagasha P, Perazella MA. ACE-I/ARB therapy prior to contrast exposure: what should the clinician do? Biomed Res Int. 2014;2014:423848]
- Why did the authors use “acute renal failure”? I think the word “acute renal failure” was an old term. KDIGO clinical practice guidelines for acute kidney injury should be cited.
Thank You. Amended
- Are there any ancestry differences in the occurrence of CI-AKI? Do the authors need to consider the participant population?
Thank You for Your observation. Ancestry differences in the occurrence in AKI are described in literature. Indeed black race is associated with a higher risk of AKI compared to non-black race; this phenomenon is thought to be due to the close interplay of clinical, socioeconomic, and genetic risk factors, which are driven by structural racial category.
We inserted this observation in the discussion: please see at pages 8-9, lines 336-338: “…Black race has an increased risk of AKI occurrence which seem to be related to direct interaction between genetic as well as clinical and social class factors [58]...”. We will certainly take this factor into account in the post-hoc analysis.
- Will the authors evaluate or adjust the amount of the iodinate CM?
Thank You for Your observation. We will evaluate for each procedure the total amount of iodinated CM. Please see page 5, lines 245-246: “…For each procedure, the anthropometric and clinical data of each patient will be collected as well as the total volume of iodinated CM used…”
- In the material and methods, what does the “50 pts” stand for? If it is the number of patients, how did the authors determine this number? Did the authors estimate based on the statistical analysis?
Thank You for this important comment. Based on our previous results [44], considering an estimated incidence of CA-AKI following primary percutaneous coronary intervention of about 20% and predicting that GSH infusion would reduce the incidence of CA-AKI of about 9% of entire population, we assumed that the enrolment of 400 patients would provide adequate power (70%) to rule out a Type I error with an alpha level of 0.05%. We have added a description of the power analysis at page 5, lines 208-212. As a consequence we implemented the number of patients to be enrolled at 400.
- Please make sure that the first time the authors use an abbreviation, it's important to spell out the full term and put the abbreviation in parentheses.
Thank You. We checked it in the text.
Reviewer 3 Report
The authors plan a double-blinded multicenter clinical trial to prove antioxidants are effective to prevent contrast-induced nephropathy in patients undergo coronary arteriogram due to STEMI. The protocol is well designed except for the rationale to decide 100 cases to investigate. The authors must calculate statistical power before set the cohort number.
Author Response
The authors plan a double-blinded multicenter clinical trial to prove antioxidants are effective to prevent contrast-induced nephropathy in patients undergo coronary arteriogram due to STEMI. The protocol is well designed except for the rationale to decide 100 cases to investigate. The authors must calculate statistical power before set the cohort number.
Thank You for Your comment. Based on our previous results [44], considering an estimated incidence of CA-AKI following pPCI of about 20% and predicting that GSH infusion would reduce the incidence of CA-AKI of about 9% of entire population, we assumed that the enrolment of 400 patients would provide adequate power (70%) to rule out a Type I error with an alpha level of 0.05%. We have added a description of the power analysis at page 5, lines 208-212. As a consequence we implemented the number of patients to be enrolled at 400.